# Italian Ryegrass as a Forage Crop for the Baltics: Opportunities and Challenges in Light of Climate Change

**DOI:** 10.3390/plants12223841

**Published:** 2023-11-13

**Authors:** Vilma Kemešytė, Gražina Statkevičiūtė, Eglė Norkevičienė, Kristina Jaškūnė

**Affiliations:** Lithuanian Research Centre of Agriculture and Forestry, Institute of Agriculture, Instituto al. 1, 58344 Akademija, Lithuania; vilma.kemesyte@lammc.lt (V.K.); grazina.statkeviciute@lammc.lt (G.S.); egle.norkeviciene@lammc.lt (E.N.)

**Keywords:** abiotic stress, dry matter yield, *Lolium multiflorum*, productivity

## Abstract

Grasslands are important for sustainable milk and meat production as well as for providing other ecosystem services. One of the most productive components of short-term grasslands is Italian ryegrass (*Lolium multiflorum* subsp. *italicum* Lam.), offering high yield, excellent feed value, and high palatability to animals but low tolerance to abiotic stress. Global climate warming opens new opportunities and could be beneficial in increasing the potential of biomass production. In this study, we aimed to assess an Italian ryegrass cultivar of Lithuanian origin, ‘Ugnė’, for productivity and yield stability, with special emphasis on their relationship with climatic factors over a period of 14 years. The average winter temperatures and total spring precipitation explained 51% of the first-cut dry matter yield (DMY) variance. Second- and third-cut DMYs were associated with average temperature only. Italian ryegrass cv. ‘Ugnė’ demonstrated the potential to produce high dry matter yields after warm winters and withstand summer drought spells under Lithuanian conditions. However, mid-to-late-summer heat waves might reduce productivity and should be taken into consideration when breeding new Italian ryegrass cultivars.

## 1. Introduction

Grasslands are valued for the provision of various ecosystem services, not only with an environmental impact but it also beneficial from an agronomic perspective as they play a major role in sustainable milk and meat production [1,2]. Grasslands are available in different environments and for different purposes, such as grazing, silage, and hay, and they are usually composed on the basis of knowledge about the adaptation, performance, and persistence of individual species and cultivars in pure stands. One of the best forage grass species in terms of yield and energy content is Italian ryegrass (*Lolium multiflorum* subsp. *italicum* Lam.), which is often used as one of the components for silage production [3,4]. It is very productive under intensive farming systems and has high palatability; thus, it is preferred by grazing animals [5]. The inclusion of Italian ryegrass in pastures, along with legumes and other monocot species, results in higher forage quality as well as high and steady production, which can be obtained in the year of sowing [6]. In addition to its direct agronomic benefits, it provides other ecosystem services that contribute to soil health [7] by reducing nitrogen and phosphorus leaching [8,9] and, thus, the pollution of water bodies [10], as well as capturing carbon [11,12,13], increasing the stability of soil aggregates [12], and, finally, increasing the diversity and density of small fauna and biota species in the soil [14,15]. Italian ryegrass is grown as a short-lived species as it does not tolerate cold winter conditions, though a study by Canadian researchers indicated that under milder winter conditions or with sufficient snow insulation, it stays in mixtures for 4 years [1,16]. In Lithuania and other Baltic countries, it is cultivated as an annual or biennial, and the length of the cultivation period depends on the environmental conditions [17]. Low tolerance to abiotic stress, such as fluctuating temperature causing repeated freezing–thawing cycles, water logging, low levels of snow insulation, etc., are the main factors making the cultivation of Italian ryegrass difficult and limiting its distribution [18]. Italian ryegrass is used as a catch and cover crop in Lithuania [19,20], but it has low recognition as a forage grass among farmers and, thus, has received little or no attention from researchers as a prospective crop for feed production.

Projected and ongoing climate change challenges the employed agricultural practices for feed production [21,22]. In the temperate region, where Lithuania is located, the main limiting factor for high crop productivity is the rather short and moderately cool vegetative season accompanied by abiotic and biotic stresses [23]. Global warming resulting in increased annual average temperatures, shifted growing seasons, and milder winters could be beneficial in the form of increased potential for biomass production [24]. On the other hand, new possibilities will be followed by new types of stresses because of the uneven distribution of precipitation [22]. However, in temperate regions, summer droughts are mild and do not compromise crop survival, which significantly affects productivity [25]. The modification of current farming systems or the development of new ones would allow us to maintain the profitability of pastures [26,27], where crop breeding can play a key role [28]. Studies of genotype x environment interactions could be used to identify optimal cultivars or species with the best adaptation and productivity in a given environment [29,30,31]. Cultivar resistance, growth capacity, and maximized resource use efficiency [32,33,34] are also traits of high importance and must be considered when breeding in future climates.

Italian ryegrass breeding in Lithuania started in 1990, and more than a decade later, it resulted in a tetraploid cultivar, ‘Ugnė’. It is characterized by a high dry matter yield and high regrowth capacity after cuts. One of the most important characteristics of this variety is its yield stability. However, high stability usually compromises high productivity; thus, a combination of these traits is what breeders should aim for.

The aim of this study was to reveal the potential of the Italian ryegrass tetraploid cultivar ‘Ugnė’ as a forage grass under Lithuanian weather conditions by assessing its agronomically important traits and putting a special focus on its productivity and stability within and among the growing seasons over a period of thirteen years.

## 2. Results

The Italian ryegrass field trial lasted for 13 years, from 2009 to 2022, except the season of 2015. The total dry matter yield (DMY) ranged from 5586 kg ha^−1^ to 14,781 kg ha^−1^; CV = 26.9% (Figure 1). ANOVA results indicated that year had a significant effect on the total DMY and DMY of each cut (*p* < 0.001). The first-cut DMY highly varied between the years (CV = 59%), from 24% to 72% of the total DMY, and had a very strong positive correlation (r = 0.95) with it. The second-cut DMY varied the least; the CV was 19%. It ranged from 49% to 22% of the total DMY (35% on average). The CV of the third cut was 50%, but it only contributed 16% of the total DMY on average. The fourth cut was only harvested in 4 out of the 13 years and had the smallest DMY, which made up only 1% to 15% of the total DMY. The last 3 years of the experiment were also the most productive and had high first- and second-cut DMYs but low third-cut DMYs (Table 1).

Regression analysis indicated that, out of all weather variables, only the winter period (WP) average temperature (T_mean_) and total precipitation during the first cut growth period (GP) had a significant effect on the first-cut DMY. The model explained 51% of yield variance (*p* < 0.05). The effects of the accumulated growth-degree days (GDD), T_mean_, total precipitation, and GP duration on the DMY of subsequent cuts were also tested. Only T_avg_ had a significant effect on the second-cut DMY (R^2^ = 0.435, *p* < 0.05) and third-cut DMY (R^2^ = 0.38, *p* < 0.05), according to the regression analysis (Figure 2). A lower T_mean_ during the growth period was associated with a higher DMY. The tested autumn weather variables did not have a statistically significant effect on the DMY.

Winter survival, spring growth, and re-growth after cuts were visually evaluated on a scale from 1 to 9, where 1 indicates no surviving plants/no growth, and 9 indicates 100% survival/very strong growth. The lowest winter survival rate was recorded in 2013 and 2014 (Figure 3a). Spring growth was good in all years, except 2014, which coincided with the poorest winter survival. The exact cause of such poor survival and growth remains unclear, as no statistically significant factors could be identified in this study. A moderate-strong correlation was found between winter survival and first-cut DMY (*rho* = 0.68, *p* < 0.05), but no significant correlation was observed between spring growth and first-cut DMY. Re-growth after cuts was the lowest in 2010 (score 5), whereas, in other growth seasons, it was 7 or higher, even though some summers experienced rather long drought spells. A moderate correlation was estimated between the re-growth scores and cumulated DMY of the second and third cuts (*rho* = 0.60, *p* < 0.05).

Crown rust and leaf spot diseases typically manifest in Lithuania towards the end of the GP and are rarely severe; thus, they do not significantly affect yield quality. In the years of the study, leaf spots began to spread in the second half of July and the beginning of August. An exception was in 2013 when the first symptoms of damage were observed only at the beginning of September. Crown rust started spreading almost a month later, between the end of August and the beginning of September. The year 2011 was exceptional, as no symptoms of either rust or leaf spot were observed (Figure 3b).

The first cut forage quality was assessed at the beginning (2009–2010) and the end (2021–2022) of the experiment. Dry matter digestibility (DMD) and water-soluble carbohydrates (WSC) exhibited minimal fluctuations (Figure 4).

The highest DMD and the lowest crude fiber (CF) contents were estimated in 2010. The DMY in 2022 was the highest for crude protein (CP) and the lowest for WSC.

## 3. Discussion

Growing concerns about worldwide climatic changes leading to new risks for sustainable forage production have prompted numerous studies to analyze and model the effects of various climatic factors on grassland productivity [24,35,36,37,38]. The primary focus has been on increasing the frequency of drought events [39,40,41,42]. However, winterkill can still pose a serious threat in Northern Europe, especially due to warm autumns and warm weather spells in winter, which result in poor plant acclimation [30,43]. Forage mixtures in Northern regions usually include timothy, meadow fescue, and perennial ryegrass [44]. Ryegrasses are known for their higher forage quality compared to other fodder grass species [16]. Italian ryegrass is even more productive than closely related perennial ryegrass [45]. However, it is less winter-hardy and can completely die out under harsh conditions, making it a spring-sown annual crop in countries like Finland [36] and Canada [46]. Farmers in regions of Europe with relatively cold winters, including Lithuania, are reluctant to include it in forage grass mixtures due to concerns about poor overwintering. Festulolium is often considered a stress-resistant alternative to ryegrasses [47]. Indeed, in this experiment, cv. ‘Ugnė’ demonstrated lower winter survival compared to festulolium in a similar trial conducted at the same location from 2013 to 2019 [31]. However, rather poor survival (score < 6) was recorded in only two out of 13 years, which was offset by excellent spring growth, with the exception of 2014. Numerous climatic factors and their complex interactions can affect winter hardiness [35,43]. Warmer autumn temperatures can delay the onset of cold acclimation until periods of very low irradiance and shortest days, resulting in reduced freezing tolerance in perennial ryegrass and timothy [48]. Some research studies show that cold tolerance in Italian ryegrass can be improved through selection under controlled conditions, but this does not necessarily translate into better field performance [49]. The relationship between climatic factors during autumn and winter and winter plant survival could not be statistically confirmed in this study. More factors, such as daylight duration and periods of freeze–thaw with ice encasement events, should possibly be taken into consideration. The winter of 2014 was mostly warm, except for a sudden drop below −12 °C in the middle of January with low snow cover. This event likely led to high winterkill and subsequent poor spring growth. On the other hand, from 2016 to 2022, winter survival was consistently high, supporting the idea that in Lithuania, a shift to warmer autumns and winters is more of a benefit for Italian ryegrass rather than a new danger to crop survival.

High herbage and/or dry matter yields are the main focus of every farmer and forage grass breeder. In this study, Italian ryegrass DMY varied greatly between years and was mainly dependent on the first cut. The highest yield in 2022 was 2.6 times higher than in the least productive season of 2013. Early spring droughts are becoming more frequent in Lithuania, which raises concerns that first-cut productivity of forage grasses might become compromised in the future. The driest spring was recorded in 2019, with only 5 mm of precipitation from the start of the growing period to the first cut, after a warm winter (T_mean_ 0 °C). In contrast, the coldest winter in the study period (T_mean_ −5.4 °C) preceded a rainy spring (140 mm total precipitation). The first-cut DMY was very similar in these two contrasting years. The highest first-cut DMY was harvested in 2022, which also had the highest spring precipitation (158 mm) and a mild winter (T_mean_ −0.4 °C). The second-highest first-cut DMY was harvested in 2020, after a rather dry spring (54 mm precipitation) and the warmest winter (T_mean_ 3.1 °C) in the study period. Accumulated spring temperatures and precipitation were found to be the main factors explaining the DMY variation in South Korea [50,51]. In another study, severe winter cold and spring drought were shown to limit the productivity of Italian ryegrass [52]. Higher spring precipitation combined with warmer winters led to higher first-cut yields in this study, suggesting that predicted warm winters in the near future may partially offset spring droughts in the region.

Accumulated precipitation did not have significant effects on the DMY of the second and third cuts, even though summer drought spells were recorded, especially in the last four years (2018–2022). The droughts in the Baltic-Nordic regions are relatively mild and, therefore, not as detrimental to crops as compared to Southern Europe [24]. Previous studies have shown that summer drought can reduce the DMY of perennial ryegrass in Lithuania [53], but the tetraploid genotypes were less susceptible. Furthermore, there may be significant variations in drought tolerance among the cultivars [42]. This study used a single tetraploid cultivar; therefore, it is unclear whether water deprivation during the summer was not severe enough to become a limiting factor for Italian ryegrass as a species or for the cultivar ‘Ugnė’ in particular. Heatwaves in the middle to end of summer in 2020 and 2021, rather than precipitation, negatively affected not only the DMY of the second and third cuts but also resulted in a decrease in forage palatability and digestibility due to a higher stem proportion. If the trend of high summer temperatures continues in the future, it might discourage farmers from considering the addition of Italian ryegrass to grazing mixtures. On the other hand, adding Italian ryegrass silage to animal diets increases forage intake [54]. Therefore, it can be an excellent choice for silage production due to its stable DMD and WSC content and high first-cut DMY. Climate change not only has a direct effect on crop and forage production but also an indirect effect by altering the forage quality through changes in nutrient availability and composition [55].

Perennial grasses play an important role in accumulating a wide range of nutrients, with each grass species in the grassland playing a different role in providing nutrients for livestock. Among these, ryegrass stands out as a source of energy due to its high carbohydrate content. This results in good digestibility throughout the entire vegetative season, even as the CF content increases as the plant matures [56]. This was confirmed in our study when the high CF content (29.1%) assessed in 2021 was counterbalanced by a high WSC content, resulting in a consistently high DMD. High DMD (69.3–80%) and WSC content (26.7–29.6%) were consistently observed throughout the study period, whereas the highest CP content was assessed in 2022, confirming a negative correlation between CP and WSC [2].

Fungal diseases, such as crown rust, can significantly reduce plant fitness, herbage yield quality, and seed yield and negatively impact seedling vigor in plants from rust-infected parents [57,58]. In this study, fungal disease symptoms were rather low, suggesting that climatic conditions are still unfavorable for the early spread of fungal infections in forage crops in Lithuania [31,33]. Very similar trends have been observed in earlier studies on festulolium and perennial ryegrass in Lithuania [31,33], as well as in Norway and other Nordic countries [23,59]. A three-year study of fungal infections in perennial ryegrass in Estonia demonstrated a significant effect of cultivar and year on rust damage in tetraploid perennial ryegrass, whereas diploid cultivars were primarily damaged by leaf spot [60]. Although fungal diseases currently cause minimal forage damage in Lithuania and can be controlled through good management practices, disease resistance should not be disregarded by grass breeders due to predicted longer and warmer vegetation periods in the future.

## 4. Materials and Methods

### 4.1. Plant Material and Experimental Design

The Lithuanian-origin tetraploid Italian ryegrass (*Lolium multiflorum* subsp. *italicum* Lam.) cultivar ‘Ugnė’ was evaluated under field conditions at the LAMMC Institute of Agriculture (55°23′ N, 23°57′ E) over a period of fourteen years. The experiment was set up in a six-field rotation, 6.52–9.75 m^2^ test plots using a randomized complete block design with three replications, with each experimental cycle consisting of a sowing year and a main harvest year. The field soil was *Endocalcari–Epihypogleyic Cambisols* (*CMg-p-w-can*), which was characterized by a sandy loam texture with a pH_KCl_ 7.3–7.0, humus content 1.9–2.2%, available P_2_O_5_ content 206–270 mg kg^−1^, and K_2_O content 101–154 mg kg^−1^ (determined using the Egner–Rim–Doming (A–L) method; Appendix A). Seeds were sown in mid-June using a Hege 80 seed drill at a depth of 2.0–2.5 cm, with a seeding rate of 25 kg of pure live seeds per hectare. Basic nitrogen, phosphorus, and potassium fertilizers were applied before sowing (NPK 10-120–180 kg ha^−1^), at the beginning of each vegetative season (N60 kg ha^−1^), and after each cut (N45 kg ha^−1^), except for the last one. The annual nitrogen fertilizer rate varied from 150 to 195 kg ha^−1^, depending on whether three or four cuts were harvested.

### 4.2. Determination of Productivity and Quality Characteristics

Productivity was determined throughout the entire vegetative season, with plots harvested 3–4 times per season depending on the meteorological conditions. Plots were harvested using a self-propelled hay mower, cutting at approximately 5 cm above ground level, and biomass was weighed in the mower’s bunker. Samples of 0.5 kg of fresh biomass were dried at 105 °C in a well-ventilated oven until a constant weight was achieved. The dried samples were used to determine the dry matter yield (DMY). Harvests were conducted when more than 50% of the plants reached the heading stage. Subsequent harvests depended on favourable or unfavourable growth conditions and were carried out at intervals of 40–60 days. Winter survival, spring growth, and regrowth after cuts were visually evaluated and scored on a scale of 1–9, where 1 represented the lowest and 9 represented the highest value of the trait. Crown rust (*Puccinia coronifera* Kleb) and spot disease (*Drechslera* spp.) damage were scored as follows: 0 = no damage, 1 = trace of disease, 2 = 5%, 3 = 10%, 4 = 20%, 5 = 30%, 6 = 40%, 7 = 50%, 8 = 75%, and 9 = more than 75% of the leaves covered with disease symptoms.

The nutritive value of the herbage was determined at the heading stage of the plants, just before the first cut. Samples were analyzed for contents of crude protein (CP), crude fiber (CF), water-soluble carbohydrates (WSC), and dry matter digestibility (DMD) using a near-infrared spectrometer NIRS-6500 (Perstorp Analytical, Silver Spring, MD, USA). Calibration equations described by Butkutė et al. [61] were employed.

### 4.3. Meteorological Data

Lithuania is situated in the nemoral zone, characterized by a cool temperate climate and a relatively short growing season of 190–195 days [62]. Meteorological data of mean, minimal and maximal temperatures, precipitation, and snow cover were recorded at the meteorological station in Akademija (55°23′ N, 23°57′ E) from 2008 to 2022. The agroclimatic indices considered were the growth period (GP), autumn hardening (AH), and winter period (WP). The growth period was assumed to start when the mean temperature T_mean_ stayed at or above 5 °C for 5 consecutive days and lasted until each harvest. Autumn hardening started when T_mean_ dropped to or below 5 °C for 5 consecutive days, ending on the first day of T_min_ ≤ −10 °C. Cold days were defined as T_min_ ≤ −15 °C. Degree days were calculated with a base temperature of T_b_ = 5 °C [35,63]. Meteorological conditions varied in terms of both precipitation and temperature during the study period (Appendix A). Although Lithuania is in a zone with excess irrigation, summer droughts are relatively frequent. The vegetative season of 2015 was dry but cool, and the lack of precipitation during the seasons of 2018, 2019, 2020, and 2021 was accompanied by high temperatures. The summer of 2022 was characterized by hot and moist weather. In contrast, the summer of 2017 was cool and moist. Particularly cold winters occurred in 2010 and 2014, whereas the winters of 2020 and 2022 were very warm. Meanwhile, the winter period of 2016–2019 featured short periods of about 3–4 weeks of cold when the temperatures dropped below −15 °C, which is critical for ryegrass.

### 4.4. Statistical Analysis

The statistical analysis was conducted in the open-source R statistical environment (version 4.3.1) [64]. Basic descriptive statistics were calculated using the R package ‘metan’ function desc_stat [65], and analysis of variance along with post hoc tests were performed using the R package ‘agricolae’ [66]. To estimate the environmental effect on DMY, ANOVA was employed, followed by a post hoc Tukey HSD test. The Kruskal–Wallis test was used to analyze the influence of the year on qualitative traits. Linear regression analysis was applied to evaluate the relationship between dry matter yield and climatic factors. The factors used for predicting the first-cut DMY included AH length (days), AH T_mean_ (°C), AH precipitation (mm), WP length, WP T_mean_, WP Cold days, WP precipitation, GP1 (from the start of GP to the first cut) length, GP1 T_mean_, GP1 precipitation, and GP1 accumulated degree days (DD). The corresponding T_mean_, length, precipitation, and accumulated DD of the growth period were used to calculate the influence of climatic factors on the second-cut and third-cut DMY. Only statistically significant effects (*p* < 0.05) are reported in the results. Spearman’s correlation coefficient(rho) was used to estimate the relationship between winter survival and spring growth scores and the first-cut DMY, as well as the relationship between cumulated second- and third-cut DMY and re-growth after-cut scores.

## 5. Conclusions

Climate change can increase forage grass productivity due to increasing winter temperatures, thereby promoting the cultivation of Italian ryegrass, which was previously considered not winter-hardy enough for the Baltic region. Field trial results over the period of 14 years demonstrated its potential to produce very high and high-quality dry matter yields after warm winters and withstand summer drought spells under Lithuanian conditions. However, mid-to-late-summer heat waves may reduce productivity and should be taken into consideration when selecting new cultivars.

## Figures and Tables

**Figure 1 plants-12-03841-f001:**
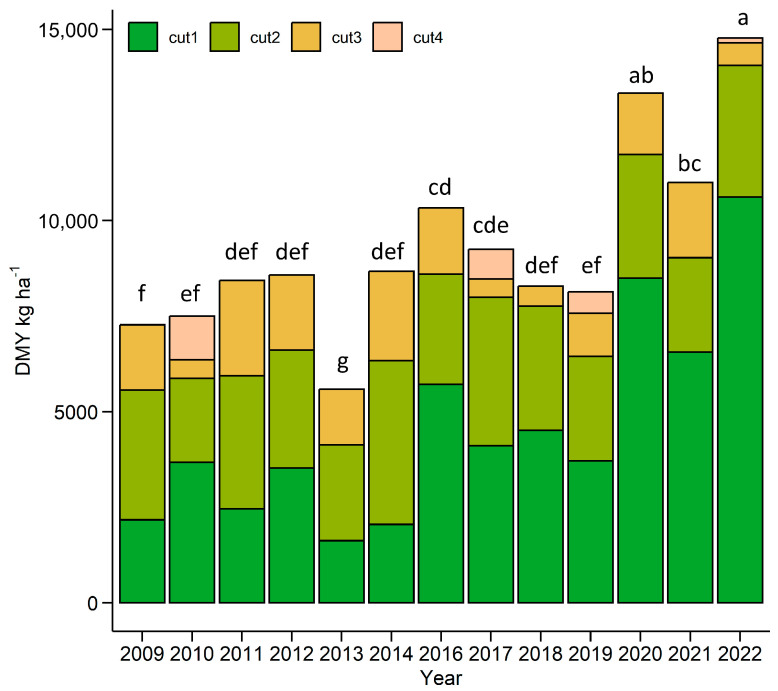
The total dry matter yield (DMY) over the period of 2009−2022. The letters indicate significant differences (Tukey’s honest significance test, *p* < 0.05).

**Figure 2 plants-12-03841-f002:**
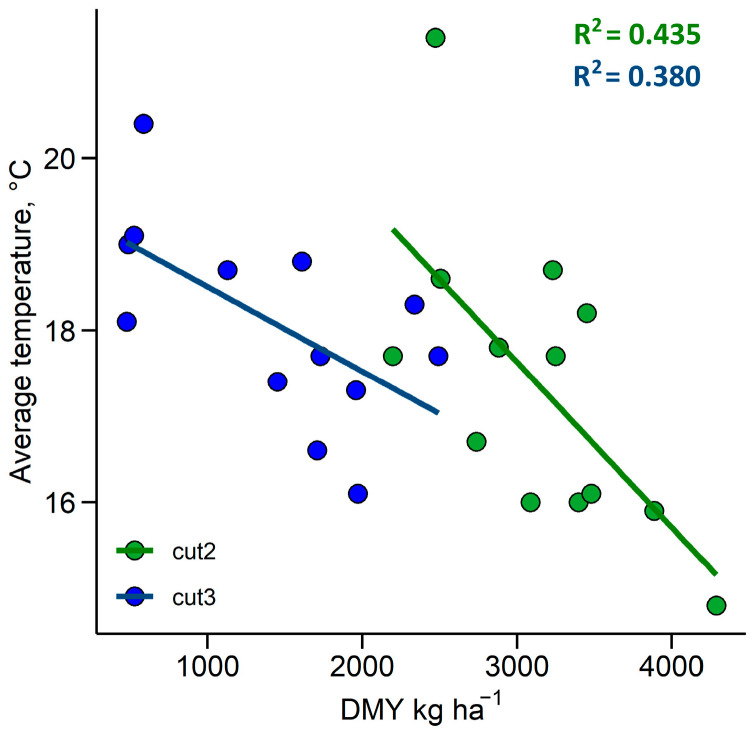
The relationship between the dry matter yield (DMY) of the second (cut2) and third (cut3) cuts and the average temperature during the corresponding growth period. The lines represent linear regression.

**Figure 3 plants-12-03841-f003:**
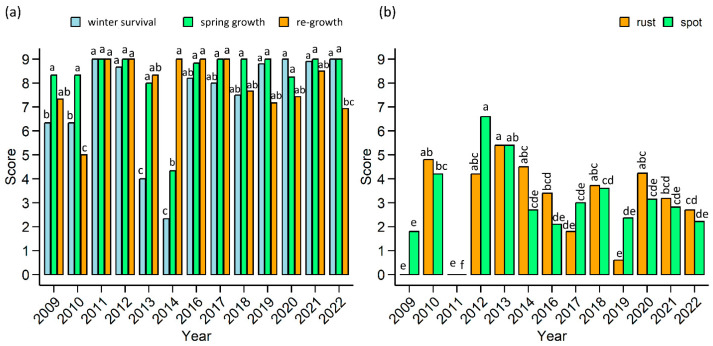
Mean scores of various traits over the period of 2009−2022. (**a**) Winter survival, spring growth, and re-growth after cuts; (**b**) crown rust and leaf spot diseases. The letters indicate significant differences between the means of each trait (Tukey’s honest significance test, *p* < 0.05).

**Figure 4 plants-12-03841-f004:**
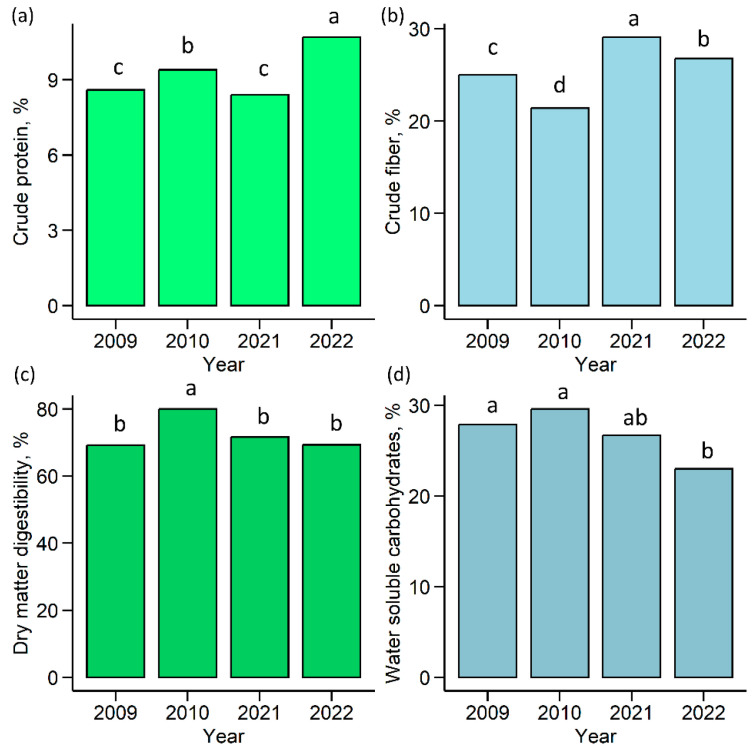
The quality of the first-cut dry matter yield. (**a**) Crude protein; (**b**) crude fiber; (**c**) dry matter digestibility; (**d**) water-soluble carbohydrates. The letters indicate significant differences (Tukey’s honest significance test, *p* < 0.05).

**Table 1 plants-12-03841-t001:** The total dry matter yield (DMY, kg ha^−1^) of each cut over the period of 2009−2022. The letters next to the DMY value indicate significant differences between the years (Tukey’s honest significance test, *p* < 0.05).

Year	Cut 1	Cut 2	Cut 3	Cut 4
2009	2172 gh	3397 abc	1708 abcd	-
2010	3678 ef	2198 c	487 e	1137 a
2011	2461 fgh	3480 abc	2492 a	-
2012	3531 efg	3086 abc	1959 abc	-
2013	1629 h	2506 c	1452 cd	-
2014	2053 h	4285 a	2337 ab	-
2016	5719 cd	2882 bc	1729 abcd	-
2017	4109 e	3887 ab	478 e	778 ab
2018	4513 de	3249 abc	525 e	-
2019	3712 ef	2737 bc	1129 de	559 ab
2020	8500 b	3231 abc	1610 bcd	-
2021	6559 c	2473 c	1972 abc	
2022	10616 a	3451 abc	586 e	127 b

## Data Availability

Data sets analyzed during the current study are available from the authors upon reasonable request.

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
