# Peer review of "Italian Ryegrass as a Forage Crop for the Baltics: Opportunities and Challenges in Light of Climate Change"

_plants, 2023, doi:10.3390/plants12223841_

Round 1
Reviewer 1 Report
Comments and Suggestions for Authors
row 45-46 mentioned factors influence agronomic practice and use, not agricultural and commercial importance;
citation is missing
row 48 feed production, especially in relation to the object of study - forage grass
row 58 varieties with improved resistance….are the other way of identifying optimal cultivar:
it is necessary to unify the terms - variety and cultivar
the sentence makes no sense – should be mentioned an approach (or better two approaches or the ways), not a variety
this statement is also at odds with the study's content, where only one single variety was tested (in a long time period and thus out of date)
row 98 0 to 9 scale was used – but 1 means no surviving plants or no growth; what then represents degree 0?
row 127 missing dot
row 129 inappropriate reference - does not deal with Italian ryegrass
row 134 missing space - ryegrasses [38]
row 135 it would be appropriate to compare the tested cultivar not with festulolium, but with another ryegrass cultivar
row 140 again – what situation is in the case of Italian ryegrass, perennial ryegrass has really a different norm of response to abiotic stress
row 156 it is nice that the authors mention the situation in South Korea, but I would assume that they will pay much more attention to the relationship of climatic factors in Lithuania and the yield in the trial plots, this is not well explained and the supplementary data is not properly used
row 164 missing space
row 166 drought tolerance is better term than resistance, it should be corrected in the text
row 176 is not sufficiently discussed, it is not linked to the results - or better - the results are not discussed at all
in general – discussion should be improved and focused on discussion of authors results
row 184 a significant insufficiency is the use of only one cultivar
row 187 why was not used the same methodology and the same parcel size?, can't it affect the results?
row 230 missing space
Comments on the Quality of English Language
minor corrections - see comments to row 58, unification of terminology - cultivar - variety, resistance - tolerance
Author Response
Dear Reviewer,
Thank you very much for your reports on reviewing the ID plants-2674607 “Italian ryegrass as a forage crop for the Baltics: opportunities and challenges in the light of climate change”. We highly appreciate the time and effort you put into reading and reviewing our submission. In the Review Report file, we describe in detail how we have addressed the issues raised and as requested, all amendments have been highlighted in blue in the manuscript.
We think that our manuscript improved through your helpful advice and thank you once again for your efforts.
Sincerely,
On behalf of the authorship,
Kristina Jaškūnė

Reviewer 2 Report
Comments and Suggestions for Authors
The manuscript is well written, but there are some shortcomings which should be removed or corrected.
line 31: due to high soluble carbohydrates content the hay making from it. ryegrass is not a good way how to preserve the forage (risk of molding or fire)
line 39 - all the mentioned benefits can be written for all perennial grasses
figure 2: R2 values should be shown also in the graph, not just in the text
line 111 - insteadof yield quality, forage quality should be written
figure 3: the CP value is very low and WSC very high - I miss the discussion to this figures, the values from NIRS are sometimes suspicious
line 175: due to heat waves and drought not just the yield but esp. digestibility, and palatability decreases in 2nd and 3rd cuts due to higher stem proportion in the forage and diseases occurence
line 182: why this chapter is not before results? Is is nonlogical
line 188: instead the term year of use, the correct term should be "the main harvest year"
line 190: without the extraction method the values of available nutrients concentration are useless
line 192: Imiss the information on the term of seeding? Was it in spring or in autumn? If in spring, was the forage yield evaluated too?
line 193: Are the figures in the brackets the application rates in kg/ha?
line 194: What was the total N rate per year? There were 3 or 4 cuts per year.
line 197: Did you evaluate just the main harvest year? What about year of establisment?
line 203: you probably mean when 50% stems were in heading stage (instead of emerged plants)
line 204: you mention 30 - 40 days interval befere the individual cuts. How did you decided for the 4th cut?
line 205: damage and were... something is missing here
line 213: in this case not just the type of spectrometer, but namely the source of calibration equations is needed
Author Response

(The authors gave the same response as above.)

Reviewer 3 Report
Comments and Suggestions for Authors
The authors conducted a field trial on Italian ryegrass, testing dry matter yield, herbage quality, and various traits of winter survival and growth over 14 years in Lithuania. The study has the potential to provide timely and relevant information, but it cannot be published in its current form. Several weaknesses in the manuscript need to be addressed before being published.
The comments/suggestions are as follows:
The title of the study is too general and doesn't accurately represent the study’s content.
The Materials and Methods section lacks some information and often is unclear.
¾ The experiment consisted of 13 years, not 14.
¾ The texture of the soil is missing.
¾ There is no information on when the trials were seeded. Please report the growing period.
¾ The soil nutrient status cannot be the same at the beginning of each growing season. Please clarify.
¾ Nitrogen content in the soil before seeding is not reported.
¾ The amount of fertilizer applied during the experimental period is not reported correctly.
¾ Since plots have been set up every year, were they seeded in the same place? Please clarify this in the text.
¾ Authors reported that the cutting interval after the first cut is 40-60 days (lines 203-204). What does it mean? If the intercutting is based on the number of days, it cannot change from 40 to 60. If it is variable, the comparison between cuts over the years cannot be made because it is affected by intercutting growth that could be markedly different (40 days regrowth vs 60 days regrowth).
¾ All the parameters and effects analyzed should be reported in the paragraph of statistical analysis. Regression analysis is not described; specify all the weather variables regressed.
Objective
The aim of the study is focused on DM yield, but several other parameters have been investigated. What about herbage quality, winter survival, and so on? They are not mentioned di in the objective of the study. Moreover, “productivity among and within seasons” is unclear. Maybe you meant growing season.
Results section
The results of ANOVA are not mentioned. I understand that mean separation tests are performed if there is a significant difference. However, the significant difference must be specified in the text or a table reporting all the p values.
The discussion of regression results is unclear and confusing. To help readers understand, I suggest you report a table with all the regression results. The readers do not know the variables that were regressed.
Descriptions that belong to Material and Methods should not be included in the Results section: lines 89-90; lines 98-100).
Results regarding Winter survival, Spring growth, and regrowth after cuts do not provide interesting information if they are not correlated to Yield. Why were correlations not performed?
The results of herbage quality were poorly discussed. There are significant differences in CP and fibers among years that merit to be discussed or at least justified.
More comments
Lines 39-41. The sentence is unclear; be more specific.
Line 59. G x E. Unclear.
Line 72. The mean total dry matter yield. The word “mean” is unnecessary and leads to confusion. Check the entire manuscript for this.
Lines 89-90 and 113-114. These statements must be moved to the M&M section.
Table 1. Specify that the letters must be read within the column.
Figure 2. The visual estimation uses the 1-9 scale, so change the Y axis from 0 - 9 to 1 - 9.
Lines 110-112. Add references.
Lines 124-133. I suggest reducing the information reported here. They are already mentioned in the Introduction.
Avoid starting the sentence with an acronym (line 221. CP, line 222 AH).
Author Response

(The authors gave the same response as above.)

Round 2
Reviewer 3 Report
Comments and Suggestions for Authors
All of my comments have been properly addressed. I have appreciated the efforts put into improving the manuscript.